# OpenReview forum: "SALE : Low-bit Estimation for Efficient Sparse Attention in Long-context LLM Prefilling"
_ICLR.cc/2026/Conference — ICLR 2026 Conference Withdrawn Submission_

### Official Review · Reviewer_jQSp · 2025-10-17

**Soundness:** 3
**Presentation:** 3
**Contribution:** 2
**Rating:** 4
**Confidence:** 5

**Summary:**

The paper proposes a training-free sparse attention method for prefilling workload of language models. It first computes the attention scores of attention sink block and sliding window block. Then using the quantized $Q$ and $K$, it calculates the attention scores for each block and compares them with those of the sink and sliding window blocks on-chip. Finally, it stores an indicator for whether each block should be computed in the attention operation, without materializing the full attention score matrix. The paper further implements customized kernels for both the attention pattern selection and the sparse attention computation. The proposed method achieves greater acceleration in attention operations while maintaining slightly better accuracy than other popular sparse attention prefill methods on standard benchmarks.

**Strengths:**

1. This paper uses fine-grained attention weight approximation via low-bit quantized $Q$, $K$, instead of using the pooling based method as adopted by many previous works. This allows more precise identification of which attention blocks are truly important.
2. The selection pass is efficiently organized. The method avoids forming the full materialization of the $N \times N$ attention matrices, whicih has the potention of achieving low prediction overhead.
3. The authors implement custom CUDA kernels (Selection-Pass and Computation-Pass), incorporate optimized tiling, use low-bit tensor core operations, and reduce DRAM I/O. Experiments are run on consumer-level GPU RTX 4090 to verify practical speedups.

**Weaknesses:**

1. **Limited accuracy improvement over baselines.** Although the proposed method achieves higher efficiency, its accuracy improvements are only marginal. On many benchmarks, the method performs comparably to or only slightly better than existing sparse attention baselines, and in several cases it does not achieve the best accuracy.
2. **Incomplete efficiency evaluation.** The efficiency analysis focuses primarily on the attention operation itself, rather than on end-to-end inference latency. Since real-world prefill performance depends on additional components such as linear layer, normalization, and residual operations, it is unclear how much overall wall-clock latency improvement the method delivers in practice. Including end-to-end latency measurements (e.g. with input lengths 16k, 32k, 64k, 128k) would make the efficiency claims more convincing. Moreover, the experiments are conducted on a consumer GPU (RTX 4090) rather than datacenter-grade hardware such as A100 or H100. For large models deployed in production, evaluation on such GPUs is necessary to validate real-world performance and scalability.

**Questions:**

1. The proposed method relies on the attention sink block and sliding window block as baselines when selecting attention patterns. While the paper focuses on language model prefill, the approach seems potentially applicable to diffusion transformers, which also use local or sliding-window attention (sliding tiles [1]). Could the authors provide some results on diffusion transformers, or comment on whether they have considered applying their method to such models?
2. Related to weakness 1, what is the accuracy result (longbench or infinitebench) when the sparsity/attention operation time is roughly the same across different baselines? This would give a clearer picture of whether the proposed method provides a genuine accuracy advantage, independent of speed improvements.
3. Related to weakness 2, could the authors provide end-to-end latency measurements for different input lengths (16k, 32k, 64k, 128k) and sparsity under configuration of the main experiment? It would also be helpful to include a breakdown of the runtime among the attention operation, linear layers, and other components to better understand the sources of speedup.
4. Related to weakness 2, could the authors evaluate the method on datacenter-grade GPUs such as A100 or H100? For large models, such hardware is typically used in practice, and it would help demonstrate the real-world scalability and efficiency of the proposed method.

[1] Fast Video Generation with Sliding Tile Attention. Peiyuan Zhang and Yongqi Chen and Runlong Su and Hangliang Ding and Ion Stoica and Zhengzhong Liu and Hao Zhang.

---

### Official Review · Reviewer_PSTc · 2025-10-24

**Soundness:** 2
**Presentation:** 2
**Contribution:** 2
**Rating:** 4
**Confidence:** 3

**Summary:**

This paper introduces a training-free sparse attention mechanism for accelerating prefill. Key points are accelerating training through quantization and evaluating importance using relative attention scores. By customizing CUDA, the latency is approximately 11% of full attention, achieving a 3.36x speedup for long-context over 64K.

**Strengths:**

1. This paper achieves more accurate importance estimation through per-head threshold calibration.
2. A lot of work has been done on kernel optimization, and the paper also introduces the optimization ideas and details in detail.

**Weaknesses:**

1. For the stability of softmax, using the relative method is a common practice. It is not particularly innovative and can be explained more clearly in the background knowledge or other sections.
2. This does not prove the necessity of the relative importance approximation, and the same effect may be achieved through its simpler selection method.

**Questions:**

1. Do I understand this correctly? The commonly used softmax stability formula is softmax(x_i) = exp(x_i - max(x)) / Σ exp(x_j - max(x)). There are two changes here:

a. max(x) is estimated using local information (I_SL).

b. Head-wise thresholding. The softmax stability formula in Flash Attention also uses per-head or even per-query max(x). What is the difference?

---

### Official Review · Reviewer_C7Yf · 2025-10-31

**Soundness:** 3
**Presentation:** 3
**Contribution:** 3
**Rating:** 4
**Confidence:** 4

**Summary:**

This paper introduces SALE (Sparse Attention via Low-bit Estimation), a novel, training-free sparse attention method designed to accelerate the computationally expensive prefilling stage of long-context LLM inference. The core problem it addresses is that existing dynamic sparse attention methods are often too "coarse-grained" to accurately estimate the attention map, leading to a suboptimal accuracy-efficiency trade-off.

SALE proposes a fine-grained, two-stage approach to solve this:

1. Low-bit Estimation: First, SALE rapidly estimates the entire attention map by computing the $QK^{T}$ product using 4-bit quantized queries and keys. This step is extremely fast on modern GPUs by leveraging low-bit Tensor Core instructions and reduced memory access.

2. Relative Importance Selection: Second, instead of using the raw estimated weights, SALE introduces a novel "Relative Attention Score" metric. Based on the observation that "sink" (initial tokens) and "local" (nearby tokens) regions consistently have high attention scores, SALE determines the importance of any given token pair by comparing its estimated weight relative to the weights in its row's sink and local regions.

**Strengths:**

1. The paper is well-written, and the proposed method is explained clearly. The motivation for the Relative Attention Score is logical, and the kernel design is well-described.

2. The accuracy-efficiency trade-off plots (Figure 4) are the most important result and clearly demonstrate that SALE is Pareto-optimal, achieving higher accuracy at the same latency or lower latency at the same accuracy than all four baselines.

3. The idea of using a very fast, low-bit (4-bit) estimation pass to generate a fine-grained mask for a more accurate sparse 8-bit computation pass is an excellent systems-aware design. It intelligently trades a small, fixed overhead (~11%) for a massive reduction in the main $O(N^2)$ computation.

**Weaknesses:**

See Question below.

**Questions:**

1. Break-even Point: The Selection-Pass overhead is 23.9% at 8K context length. At approximately what sequence length does SALE's total latency (Selection-Pass + Computation-Pass) "break even" with the baseline FlashAttention2? It seems plausible that for contexts under 8K, SALE could be slower.

2. Robustness of Calibration: The per-head thresholds are calibrated using 5 samples from the InfiniteBench Retrieve.KV task. How robust are these thresholds? If you use thresholds calibrated on Retrieve.KV but evaluate on LongBench (or vice-versa), does the accuracy-efficiency trade-off degrade compared to a task-specific calibration?

3. Alternative to "Relative Score": The "Relative Attention Score"  is a key contribution. Did you also experiment with a simpler metric, for example, just using the 4-bit estimated weights to find a global top-k percentage of blocks (not tokens)? I am curious if the added complexity of the sink/local comparison is essential for the high accuracy.

4. More evaluation results. NIAH only test to 128K. I would like to see 1M context length. Besides, RULER is not evaluated which is quite common in long-context LLM benchmarks.

5. Add experiment for H100, B200 etc datacenter-level GPUs, instead of 4090 & 5090 only. I would like to see the performance of SALE with fewer CUDA kernels

---

### Note · Authors · 2025-12-01

I have read and agree with the venue's withdrawal policy on behalf of myself and my co-authors.